# A Star Sensor On-Orbit Calibration Method Based on Singular Value Decomposition

**DOI:** 10.3390/s19153301

**Published:** 2019-07-26

**Authors:** Liang Wu, Qian Xu, Janne Heikkilä, Zijun Zhao, Liwei Liu, Yali Niu

**Affiliations:** 1Department of Computer Science and Engineering, Changchun University of Technology, Changchun 130012, China; 2Center for Machine Vision and Signal Analysis, University of Oulu, 90014 Oulu, Finland

**Keywords:** star sensor, calibration, on-orbit, singular-value decomposition, extended Kalman filter

## Abstract

The navigation accuracy of a star sensor depends on the estimation accuracy of its optical parameters, and so, the parameters should be updated in real time to obtain the best performance. Current on-orbit calibration methods for star sensors mainly rely on the angular distance between stars, and few studies have been devoted to seeking new calibration references. In this paper, an on-orbit calibration method using singular values as the calibration reference is introduced and studied. Firstly, the camera model of the star sensor is presented. Then, on the basis of the invariance of the singular values under coordinate transformation, an on-orbit calibration method based on the singular-value decomposition (SVD) method is proposed. By means of observability analysis, an optimal model of the star combinations for calibration is explored. According to the physical interpretation of the singular-value decomposition of the star vector matrix, the singular-value selection for calibration is discussed. Finally, to demonstrate the performance of the SVD method, simulation calibrations are conducted by both the SVD method and the conventional angular distance-based method. The results show that the accuracy and convergence speed of both methods are similar; however, the computational cost of the SVD method is heavily reduced. Furthermore, a field experiment is conducted to verify the feasibility of the SVD method. Therefore, the SVD method performs well in the calibration of star sensors, and in particular, it is suitable for star sensors with limited computing resources.

## 1. Introduction

A star sensor is a navigation system that obtains the attitude information of the carrier by the observation of stars. They are the most accurate optical attitude sensors, at present [1]. Due to their high navigation accuracy, strong autonomy, and lack of cumulative error, they are favored in the aerospace industry [2,3]. As the “eyes” of the spacecraft, the accuracy of the star sensor determines the performance of the spacecraft directly. The star sensor is an optical device whose accuracy depends on the imaging quality and the accuracy of the optical parameters, including focal length, principal point, and distortion [4,5]. Therefore, calibration is one of the key technologies for a star sensor.

The calibration methods of star sensors can be divided into two categories: ground-based calibration and on-orbit calibration. Generally, multi-frame observations should be accumulated for the ground-based calibration method; hence, this method often relies on a fixed platform and complex experimental installations [6,7], and the cost of ground-based calibration is high. Additionally, as the working environment of a star sensor is different from the calibration environment, the parameters may change in orbit [8,9]. In most cases, on-orbit calibration is carried out with observed data during operation, such that the parameters can be updated in real time and the accuracy of the star sensor can be maintained.

The on-orbit calibration for star sensors was first studied by Junkins and his research group from Texas A&M University in 2001. According to the fact that the angular distance between stars is constant under the rotation transformation, Samaan [10] proposed an on-orbit calibration method based on the cosine residuals of the angular distances, and the detailed on-orbit calibration procedure was presented. Further studies were carried out by other members of the team; Singla [11] evaluated the performance of the calibration method based on angular distances; Griffith [12] explored approaches for the on-orbit calibration of higher order focal plane distorting effects; Woodbury [13] proposed an on-orbit calibration method based on the sine of angular distance, instead of the cosine; Liu [14] and Shen [15] concentrated on the improvement of the sequential estimation method; and Enright [16,17] suggested using the angular distances directly and adopting a new camera model for calibration. Although considerable studies have been done for the on-orbit calibration of star sensors, almost all of these were related to angular distances.

The singular-value decomposition (SVD) method was originally used for the attitude determination in a star sensor by Markley [18]. The invariant singular values of stars were applied for star identification in star sensors by Juang [19]. More characteristics of the SVD for star sensors were studied by Yin [20,21]. On the basis of these works, we find that the invariant singular value is an appropriate reference for the on-orbit calibration of star sensors, and an on-orbit calibration method based on the singular-value decomposition is proposed in this paper.

The remainder of this paper is organized as follows. The camera model to be calibrated in this paper is introduced in Section 2. In Section 3, the principle of the invariant singular values of star sensors is presented, and an SVD on-orbit calibration method with the extended Kalman filter (EKF) is proposed. Furthermore, taking observability as the criterion, further studies on the star combination models and the performance of different singular values are discussed. In Section 4, both simulation and field experiments are conducted to evaluate and verify the performance of the proposed calibration method. Finally, some conclusions are drawn.

## 2. Camera Model

It is generally accepted that a conventional camera model includes three parts: extrinsic parameters, intrinsic parameters, and distortion parameters [22,23]. Notably, a star sensor is mainly used to calculate the attitude or the rigid rotation between the camera coordinate and the inertial coordinate, so there is no need to solve for the extrinsic parameters in the calibration. Neglecting the extrinsic parameters, the coupling effect between intrinsic and extrinsic parameters can be avoided to improve the calibration accuracy. Therefore, in this paper, the camera model of a star sensor only consists of intrinsic and distortion parameters.

The reference frames of the camera model involve three coordinate frames: the camera frame, the physical image frame, and the image frame, as shown in Figure 1. Figure 1a shows the relationship between the camera frame and the physical image frame, and Figure 1b shows the relationship between the physical image frame and the image frame. These reference frames are defined based on the pinhole camera model, where *O-XYZ* is the camera reference frame, the origin *O* of the camera frame is the projection center, and the *Z*-axis is the optical axis. The physical image frame *o’-^th^ u’v’* is related to the detector plane. This frame is parallel to the *XOY* plane of the camera frame; the intersection of this plane and the optical axis is the principal point *o’*; and the *u’*- and *v’*-axes are parallel to the *X*- and *Y*-axes of the camera frame, respectively. The distance between these two planes along the optical axis is the focal length *f*. The *u*- and *v*-axes of the image reference frame *o-uv* (in Figure 1b) are parallel to the *u’*- and *v’*-axes of the physical image frame, respectively, and the origin of the image reference frame is located in the top left of the image; so, the coordinates of the principal point in the image frame are ${\left[u_{0},v_{0}\right]}^{T}$.

Let $w={\left[X,Y,Z\right]}^{T}$ be an arbitrary unit star vector [16] with respect to the camera reference frame, and its projection on the image frame is $p={\left[u,v\right]}^{T}$. The perspective projection relationship between ***w*** and ***p*** can be represented as [22,24]:(1)$$\left[\begin{array}{c} u\\ v\\ 1 \end{array}\right]=\frac{1}{z}\left[\begin{array}{ccc} f_{u} & 0 & u_{0}\\ 0 & f_{v} & v_{0}\\ 0 & 0 & 1 \end{array}\right]\left[\begin{array}{c} X\\ Y\\ Z \end{array}\right],$$
where ${\left[u,v,1\right]}^{T}$ are the homogeneous coordinates of the point ***p***, and $f_{u}$ and $f_{v}$ are:(2)$$f_{u}=\frac{f}{D_{u}},$$
(3)$$f_{v}=\frac{f}{D_{v}},$$
where *f* is the focal length and $D_{u}$ and $D_{v}$ are the horizontal and vertical pixel sizes, respectively. The relationship between the factors $D_{u}$ and $D_{v}$ is:(4)$$D_{v}=sD_{u},$$
where *s* is the aspect ratio (the height of a pixel compared to the width of that pixel) [22]. Normally, the reference values of $D_{u}$ and $D_{v}$ are given in the datasheet of the camera. From Equations (2)–(4), $f_{u}$ and $f_{v}$ are linearly dependent on the parameters *s* and *f*, and so, $u_{0}$, $v_{0}$, *s*, and *f* are the intrinsic parameters to be calibrated.

As a result of the imperfection of the camera lenses, distortion should be considered in the camera model. Lens distortion is usually expressed as:(5)$$\left\{\begin{array}{c} u_{d}=u+\delta_{u}\left(u,v\right)\\ v_{d}=v+\delta_{v}\left(u,v\right) \end{array}\right.,$$
where *u* and *v* are the distortion-free coordinates in Equation (Equation 5), $u_{d}$ and $v_{d}$ are the corresponding coordinates with distortion, and $\delta_{u}\left(u,v\right)$ and $\delta_{v}\left(u,v\right)$ are the distortions in the *u* and *v* directions, respectively. It has been shown that higher order distortions may cause numerical instability [25], so here, we consider only the first- and second-order radial distortion:(6)$$\left\{\begin{array}{c} \delta_{u}\left(u,v\right)=u\left(k_{1}r^{2}+k_{2}r^{4}\right)\\ \delta_{v}\left(u,v\right)=v\left(k_{1}r^{2}+k_{2}r^{4}\right) \end{array}\right.,$$
where $k_{1}$ and $k_{2}$ are the distortion coefficients of the radial distortion, and $r^{2}$ is defined as [24]:(7)$$r^{2}=\frac{{\left(u-u_{0}\right)}^{2}}{f_{u}^{2}}+\frac{{\left(v-v_{0}\right)}^{2}}{f_{v}^{2}}.$$

With the model above, given the position of the observed star centroid in the image frame $p_{d}={\left[u_{d},v_{d}\right]}^{T}$, we can calculate the corresponding unit star vector in the camera frame $w={\left[X,Y,Z\right]}^{T}$. We express the relationship as:(8)$$w=F(s,f,u_{0},v_{0},k_{1},k_{2},p_{d}),$$
where *F*(·) is the back-projection function with distortion.

## 3. Calibration Model Based on Singular-Value Decomposition

### 3.1. Calibration Reference: Singular Values

This calibration method is based on the invariant characteristic of the singular-value decomposition of the star sensor. The principle is presented as follows [19].

#### 3.1.1. Singular-Value Decomposition of the Star Sensor

Define $w_{i}$ as an observed star vector in the camera frame and $v_{i}$ as a guide star vector in the inertial frame. The transformation between these two frames is:(9)W=CV,
where ***W*** and ***V*** are the column vector matrices:(10)$$W={\left[\begin{array}{cccc} w_{1} & w_{2} & \cdots & w_{N} \end{array}\right]}_{3\times N},$$
(11)$$V={\left[\begin{array}{cccc} v_{1} & v_{2} & \cdots & v_{N} \end{array}\right]}_{3\times N},$$
and ***C*** is the direct cosine matrix (DCM), which indicates the transformation from the inertial frame to the camera frame. Hence, ***C*** is an orthogonal matrix.

The SVD is a general decomposition method for matrices. With the SVD, the matrices ***W*** and ***V*** can be decomposed into:(12)$$W=P_{w}\Sigma_{w}Q_{w}^{T}=\sum_{i=1}^{3}p_{wi}\sigma_{wi}q_{wi}^{T},\;\mathrm{and}$$
(13)$$V=P_{v}\Sigma_{v}Q_{v}^{T}=\sum_{i=1}^{3}p_{vi}\sigma_{vi}q_{vi}^{T},$$
where $P_{w}$ and $P_{v}$ are 3×3 orthogonal matrices of left singular vectors $p_{wi}$ and $p_{vi}$ (i = 1,2,3), $Q_{w}$ and $Q_{v}$ are N×N orthogonal matrices of right singular vectors $q_{wi}$ and $q_{vi}$ (i=1,2,⋯, N), and $\Sigma_{w}$ and $\Sigma_{v}$ are 3×N diagonal matrices where the diagonal elements are the singular values $\sigma_{wi}$ and $\sigma_{vi}$ (i=1,2,3) of ***W*** and ***V***, respectively. For no less than three stars in the field of view (FOV), there are three non-zero singular values, and the SVD is unique. The singular values have the following property.

#### 3.1.2. Invariant Singular Values

Post-multiplying Equation (Equation 9) by $W^{T}$, we obtain:(14)$${\mathrm{WW}}^{T}={\mathrm{CVV}}^{T}C^{T}.$$

Substituting Equations (12) and (13) into Equation (Equation 14) yields:(15)$$\begin{array}{cc} {\mathrm{WW}}^{T} & =P_{w}\Sigma_{w}Q_{w}^{T}Q_{w}\Sigma_{w}^{T}P_{w}^{T}=P_{w}S_{w}P_{w}^{T}\\ & ={\mathrm{CVV}}^{T}C^{T}=CP_{v}\Sigma_{v}Q_{v}^{T}Q_{v}\Sigma_{v}^{T}P_{v}^{T}C^{T}={\mathrm{CP}}_{v}S_{v}P_{v}^{T}C^{T} \end{array},$$
or:(16)$$P_{w}S_{w}P_{w}^{T}={\mathrm{CP}}_{v}S_{v}P_{v}^{T}C^{T},$$
where $S_{w}$ and $S_{v}$ are diagonal matrices with eigenvalues $\sigma_{wi}^{2}$ and $\sigma_{vi}^{2}$ (i=1,2,3) of ${\mathrm{WW}}^{T}$ and ${\mathrm{VV}}^{T}$, respectively. As ***C*** is an orthogonal matrix, Equation (Equation 14) is a similarity transformation. Thus, the eigenvalues of the ${\mathrm{WW}}^{T}$ and ${\mathrm{VV}}^{T}$ are equal; namely,

(17)$$\sigma_{wi}^{2}=\sigma_{vi}^{2},i=1,2,3.$$

Meanwhile, the positive singular values of ***W*** and ***V*** are equal:(18)$$\sigma_{wi}=\sigma_{vi},i=1,2,3.$$

Therefore, we can conclude that the singular values of the star vectors remain constant under coordinate transformation.

### 3.2. On-Orbit Calibration Method Based on Singular-Value Decomposition

The singular values can be calculated using the data obtained by the star sensor itself, and the invariance of the singular values provides a new way to calibrate the optical parameters of star sensors in orbit.

Supposing that SV(·) indicates the singular value-solving operator, Equation (Equation 18) can be presented as:(19)$$\sigma_{vi}=SV\left(V\right)=\sigma_{wi}=SV\left(W\right),i=1,2,3.$$

Substituting Equation (Equation 8) into Equation (Equation 19) yields:(20)$$\sigma_{vi}=\sigma_{wi}=SV\left(V\right)=SV\left(F\left(s,f,u_{0},v_{0},k_{1},k_{2},P_{d}\right)\right),i=1,2,3,$$
where $P_{d}=\left[P_{d1},P_{d2}\cdots P_{dN}\right]$ is the collection of observed star coordinates in the image frame.

After star identification, the observed star coordinates $P_{d}$ and the corresponding star vectors ***V*** in the star catalog are matched with each other. Thus, according to Equation (Equation 20), the singular values can be obtained by the star vectors ***V***, and they, also, can be calculated by the camera parameters and the observed star coordinates $P_{d}$. The accuracy of the star catalog is very high, and the singular values calculated by ***V*** have good precision. Obviously, Equation (Equation 20) is suitable for the measurement equation.

The measurement equation is non-linear, and thus, we estimate the camera parameters based on the extended Kalman filter (EKF) [26]. The state transition and measurement models are:(21)$$x_{k}=I_{6\times 6}\cdot x_{k-1}\;\mathrm{and}$$
(22)$$z_{k}=h\left(x_{k}\right)+n_{c},i=1,2,3,$$
where $x_{k}={\left[s,f,u_{0},v_{0},k_{1},k_{2}\right]}^{T}$, and $x_{k}$ and $x_{k-1}$ are the state parameters of the star images marked *k* and *k*-1, respectively. Suppose that the camera parameters are constant, and let $I_{6\times 6}$ be an identity matrix. Then, $z_{k}={\left[\sigma_{1},\sigma_{2},\sigma_{3}\right]}^{T}$ can be calculated by the guide star vectors ***V***; $h(x_{k})$ is a simplified representation of $SV(F\left(s,f,u_{0},v_{0},k_{1},k_{2},P_{d}\right))$; and $n_{c}$ represents the measurement error caused by the noise. Starting with initial estimates of the noise covariance $P_{0}$ and the parameters $x_{0}$, which may be obtained from the ground calibration, we process the calibration frame-by-frame. For the *k*^th^ star image, the EKF prediction equations are:(23)$$x_{k}^{-}=x_{k-1}\;\mathrm{and}$$
(24)$$P_{k}^{-}=P_{k-1}+Q,$$
where ***Q*** is the covariance matrix of the prior estimation error. In this model, the parameters are constant, and so, the theoretical ***Q*** is a null matrix. However, if this is so, the estimator is just a sequential least squaresestimator, and the artificial process noise is a common technique for forgetting old measurements.

The EKF update equations are:(25)$$K_{k}=P_{k}^{-}H_{k}^{T}{\left(H_{k}P_{k}^{-}H_{k}^{T}+R\right)}^{-1},$$
(26)$$x_{k}=x_{k}^{-}+K_{k}\left(z_{k}-h\left(x_{k}^{-}\right)\right),$$
(27)$$P_{k}=\left(I-K_{k}H_{k}\right)P_{k}^{-},$$
where ***R*** is the covariance matrix of measurement noise and $H_{k}$ is the Jacobian matrix of the measurement
(28)$$H_{k}=\frac{\partial h}{\partial x}{|}_{x_{k}^{-}}.$$

The model of the *h* is complex, so we adopt numerical differentiation to calculate the Jacobian matrix.

### 3.3. Further Study on the SVD Calibration Model

Concerning Equation (Equation 28), if all the stars in the FOV are used to calculate the singular values, it is easy to find that there are only three rows in the Jacobian matrix $H_{k}$. The order of $H_{k}$ is less than the number of the state parameters, so the observability of the estimator is bad. In order to solve this problem, the stars should be put into several groups, such that the order of $H_{k}$ can satisfy the estimator. There are many different combinations of the stars, however; how to choose the proper combination is discussed below.

#### 3.3.1. Observability Analysis

Observability is an indicator to evaluate the feasibility of the system; namely, with different models, the same input derivation may cause different output derivations. If the magnitude of the output derivation is bigger, then the observability is better and the system is more feasible, and vice versa. In this section, we define the observability as the infimum of the output derivation [27].

According to the definition of the observability, we can obtain:(29)$$\delta z_{k}=H_{k}W\delta x,$$
where δx is the input derivation vector with the same derivation for all the parameters, $\delta z_{k}$ is the output derivation, and $H_{k}$ is the Jacobian matrix. As the precision of different parameters varies considerably in practice, this means δx cannot represent the actual derivation of different parameters. Thus, δx should be weighted according to the magnitudes of different precision, where ***W*** is the diagonal weighting matrix whose elements are the typical precision of the parameters (see Table 1), according to the star sensor in the field experiment. Hence, the observability matrix is:(30)$$H_{k}^{\prime }=H_{k}W.$$

Here, the SVD is used again. With the SVD of the observability matrix, Equation (Equation 29) can be represented as:(31)$$\delta z_{k}=P_{k}\Sigma_{k}Q_{k}\delta x,$$
where $P_{k}$ and $Q_{k}$ are orthogonal matrices of left singular vectors and right singular vectors, respectively. To make sure that $H_{k}^{\prime }$ is observable, $\Sigma_{k}$ is defined as a 6×N diagonal matrix where the diagonal elements are the non-zero singular values $\sigma_{i}$ (i=1∼6).

As $P_{k}$ and $Q_{k}$ are orthogonal matrices, we can calculate:(32)$${∥\delta z_{k}∥}_{2}^{2}=\sum_{i=1}^{6}\sigma_{i}^{2}{∥\delta x∥}_{2}^{2},$$
where ${∥\delta x∥}_{2}$ and ${∥\delta z_{k}∥}_{2}$ are the two-norms of δx and $\delta z_{k}$, respectively.

The infimum of the output derivation is:(33)$${∥\delta z_{k}∥}_{2}^{2}\geq 6\sigma_{min}^{2}{∥\delta x∥}_{2}^{2},$$
where $\sigma_{min}$ is the minimum singular value (MSV) of the observability matrix. It is obvious that, with larger $\sigma_{min}$, the infimum of the output derivation is bigger and the observability is better. Therefore, we adopt $\sigma_{min}$ as the indicator to evaluate the performance of the system and find a suitable calibration model.

#### 3.3.2. Star Combination Models

With regards to no less than three stars, there are three non-zero singular values, and so, the combination can be formed by three stars, four stars, and so on. If all the combinations are used to constitute the measurement models, the computational cost is huge. In addition, the combinations are not independent, and the information for calibration does not increase linearly with the number of combinations. Here, we discuss several models based on the different number of stars in the combination.

Supposing there are *N* stars in the FOV, we set up four models, as follows.

Model-1: As shown in Figure 2a, a combination is constituted by three stars, such as 1-2-3, 2-3-4, 3-4-5, ⋯.

Model-2: As shown in Figure 2b, a combination is constituted by ⌊N/2⌋ stars (where ⌊⌋ represents rounding down).

Model-3: As shown in Figure 2c, a combination is constituted by *N-1* stars.

Model-4: As shown in Figure 2d, differing from the other models, this model is constituted by various numbers of star combinations, such as 1-2-3, 1-2-3-4, 1-2-3-4-5, ⋯.

The observability analysis of these models is shown in Figure 3, where the MSVs of the Jacobian matrix of each model are calculated frame-by-frame. The MSVs of Model-1, Model-2, and Model-3 interweave with each other, and thus, we suspect that, for different star images, combinations of different star numbers have different observabilities. The MSVs of Model-4 are bigger, in most cases, which means that the observability of Model-4 is better than the others (this may be because Model-4 combines the advantages of various star-number combinations), and so, we take Model-4 as the optimal combination model. This is only a rough discussion of the combination models; further studies on this problem are still underway.

#### 3.3.3. Singular Value Selection

During the research, one unanticipated finding was that the sensitivities of the three singular values were quite different, as shown in Figure 4: $\sigma_{1}$, $\sigma_{2}$, and $\sigma_{3}$ are three singular values with descending order, and the observability of $\sigma_{1}$ is worse than $\sigma_{2}$ and $\sigma_{3}$. Thus, we should consider whether all three singular values are suitable for the estimation of the camera parameters, and we discuss this question with respect to the physical interpretation of the SVD of the star vector matrix ***W*** (see Figure 5).

The black vector depicts the star vector $w_{i}$. According to the definition of the SVD, the left singular vectors $p_{wi}$ (*i* = 1, 2, 3) are unit vectors, so the square of the maximum singular value of ***W*** is:(34)$$\sigma_{1}^{2}=\underset{∥p_{max}∥=1}{max}∥{\mathrm{WW}}^{T}p_{max}∥=\underset{∥p_{max}∥=1}{max}∥\sum_{i=1}^{N}w_{i}w_{i}^{T}p_{max}∥,$$
where $p_{max}$ represents the left singular vector associated with the maximum singular value $\sigma_{1}$.

Equation (Equation 34) illustrates that $p_{max}$ is the vector that maximizes the sum of projections of each star vector $w_{i}$ onto $p_{max}$. In the same way, $p_{min}$, associated with the minimum singular value $\sigma_{3}$, is the vector that minimizes the sum of projections of each star vector $w_{i}$ onto $p_{min}$. Then, $p_{middle}$, which is associated with the intermediate singular value, is perpendicular to the plane generated by $p_{max}$ and $p_{min}$. An illustration of $p_{max}$, $p_{middle}$, and $p_{min}$ is shown in Figure 5.

As $p_{max}$ and $w_{i}$ are unit vectors, Equation (Equation 34) can be rewritten as:(35)$$\sigma_{1}^{2}=max\sum_{i=1}^{N}∥\cos\alpha_{i}∥,$$
where $\alpha_{i}$ is the angle between $w_{i}$ and $p_{max}$, which represents the projection of the star vector $w_{i}$ onto $p_{max}$. Therefore, the derivation of $\sigma_{1}$ is the magnitude of $\sin\alpha_{i}$. As the plane generated by $p_{middle}$ and $p_{min}$ is perpendicular to $p_{max}$, $\sigma_{2}$ and $\sigma_{3}$ are proportional to $\sin\alpha_{i}$, and the derivations of $\sigma_{2}$ and $\sigma_{3}$ are on the magnitude of $\cos\alpha_{i}$. With regards to a small FOV, $\sin\alpha_{i}$ is smaller than $\cos\alpha_{i}$, and so, the derivation of $\sigma_{1}$ is smaller than $\sigma_{2}$ and $\sigma_{3}$, which is in line with the observability analysis. As a result, in order to reduce the calculation, we can employ $\sigma_{2}$ and $\sigma_{3}$ for the measurements. As shown in Figure 5, the status of $\sigma_{2}$ and $\sigma_{3}$ is similar. The values of $\sigma_{2}$ and $\sigma_{3}$ are the sum of the projections of $\sin\alpha_{i}$ onto the $p_{middle}$ and $p_{min}$ axes, respectively. Depending on the position of the two axes, the observabilities of $\sigma_{2}$ or $\sigma_{3}$ may be large or small, resulting in an unstable effect when using $\sigma_{2}$ or $\sigma_{3}$ alone.

## 4. Results and Discussion

The simulation experiment and field experiments were conducted to verify and analyze the SVD on-orbit calibration method.

### 4.1. Simulation Experiments

Star data were simulated, with a $19.14^{\circ }\times 11.18^{\circ }$ FOV, for a 1920×1080 pixel-array star sensor at a 2-Hz update rate. The other simulation parameters are listed in Table 1. The simulation data consisted of three datasets: a set of 3D star vectors in the inertial frame $S_{3D}$, a set of corresponding 2D star coordinates in the image frame $S_{2D}$, and the set of 2D star coordinates with normally-distributed noise (with a standard deviation of 0.5 pixel), which is referred to as $S_{2D}^{\prime }$. For each experiment, $S_{3D}$, $S_{2D}$, and $S_{2D}^{\prime }$ of 2500 images were simulated. These images were divided into two groups, where the first 2400 images were used for calibration (according to the experiment, we found that 2400 images could ensure the convergence of calibration), and the last 100 images were used to evaluate the performance of the calibration. The calibration programs were written in MATLAB and run on an Intel Core i5-8300H CPU.

In order to evaluate the calibration method fully, the evaluation criteria are presented as follows. As the residual of a singular value cannot directly reflect the influence of calibration on the star sensor, the residual of angular distance between stars, which is normally used to evaluate the performance of on-orbit calibration, is adopted in this paper. Thus, two criteria to evaluate the calibration methods were employed:

Criterion A: This criterion is based on the simulation data without errors. With the estimation of camera parameters, $S_{2D}$ can be back-projected to the camera frame, and the angular distances between them are calculated. The residual errors are obtained by these angular distances and the corresponding angular distances calculated by $S_{3D}$. Then, the root-mean-squared error (RMSE) of the angular distances ($A|\delta_{RMSE}$) in each star image is calculated. The mean value ($A|\mu_{\delta_{RMSE}}$) and the standard deviation ($A|\sigma_{\delta_{RMSE}}$) of $A|\delta_{RMSE}$ in the last 100 images are used as the evaluation indices of Criterion A.

Criterion B: The difference between this criterion and Criterion A is that this model uses the noisy 2D simulation data $S_{2D}^{\prime }$ to calculate the RMSE of the angular distances ($B|\delta_{RMSE}$) in each star image. The evaluation indexes of Criterion B are presented as the mean value ($B|\mu_{\delta_{RMSE}}$) and the standard deviation ($B|\sigma_{\delta_{RMSE}}$) of $B|\delta_{RMSE}$ in the last 100 images.

Therefore, $A|\mu_{\delta_{RMSE}}$ and $A|\sigma_{\delta_{RMSE}}$ are only related to the estimation errors of camera parameters, and these evaluation indexes can only be used in the simulation. Furthermore, $B|\mu_{\delta_{RMSE}}$ and $B|\sigma_{\delta_{RMSE}}$ are related to both the estimation errors of camera parameters and the centroid noise.

Test 1. Performance of different models:

The aim of this simulation test was to evaluate the performance of the star combination models presented in Section 3.3.2. A star catalog was formed by taking stars, up to a visual magnitude of 5.5, from the Tycho-2 catalog. The initial values were set as *s*= 1, *f*= 15.5 mm, $u_{0}$= 960, $v_{0}$= 540, $k_{1}$= 0, and $k_{2}$= 0. The calibration results are listed in Table 2.

The derivation of the parameters shown in Table 2 is expressed as fractional changes, due to the different scales of the parameters. The results showed that the parameters *s*, $u_{0}$, $v_{0}$, and $k_{2}$ obtained by Model-2 and Model-3 had better accuracy than Model-1; *f* obtained by Model-1 and Model-2 had better accuracy than Model-3; and $k_{1}$ obtained by Model-1 had better accuracy than Model-2 and Model-3. On the basis of these results, we conclude that, for different parameters, each of the first three models had advantages and disadvantages. However, the precision of the parameters calibrated by Model-4 was always at an intermediate level, which means that Model-4 combined the properties of the other combination models. With regard to the error evaluation Criterion A, the performance of Model-4 was better than the others. This is consistent with the analysis in Section 3.3.2, so we used Model-4 for the other experiments.

Test 2. Performance of different singular values:

The simulation conditions were the same as in Test 1. In this test, all combinations of the singular values were used to construct the Jacobian matrix, and the calibration results are as follows.

According to the results in Table 3, the three best combinations were $\left\{\sigma_{1},\sigma_{2},\sigma_{3}\right\}$, $\left\{\sigma_{2},\sigma_{3}\right\}$, and $\left\{\sigma_{2}\right\}$. This result indicates a good agreement with the analysis in Section 3.3.3; namely, the observabilities of $\sigma_{2}$ and $\sigma_{3}$ were better than that of $\sigma_{1}$. Particularly, in this test, the observability of the Jacobian matrix related to $\sigma_{2}$ was better than $\sigma_{3}$, so the calibration results achieved only by $\sigma_{2}$ were good enough. Through several experiments, however, we found that the performance of $\left\{\sigma_{2}\right\}$ was unstable, and in this experiment, the relatively high standard deviation $A|\sigma_{\delta_{RMSE}}$ gave some indication of the stability problem. This may be due to some information in $\sigma_{1}$ and $\sigma_{3}$ that could help with calibration, and so, the combinations $\left\{\sigma_{1},\sigma_{2},\sigma_{3}\right\}$ and $\left\{\sigma_{2},\sigma_{3}\right\}$ are recommended. The calibration precisions of $\left\{\sigma_{1},\sigma_{2},\sigma_{3}\right\}$ and $\left\{\sigma_{2},\sigma_{3}\right\}$ were similar, yet the average calibration time per frame of $\left\{\sigma_{2},\sigma_{3}\right\}$ was shorter than that of $\left\{\sigma_{1},\sigma_{2},\sigma_{3}\right\}$, which means that the computation of $\left\{\sigma_{2},\sigma_{3}\right\}$ was smaller than $\left\{\sigma_{1},\sigma_{2},\sigma_{3}\right\}$. Thus, the combination $\left\{\sigma_{2},\sigma_{3}\right\}$ was adopted for the following tests.

Test 3. Comparison with the calibration method based on the angular distance:

The calibration method based on the angular distance (AD) method is widely used for the on-orbit calibration of star sensors, so we further evaluated the performance of the proposed method by comparing it with the AD method.

We set up three experiment groups, where the difference of the groups was the limiting visual magnitude of the star sensor, which led to different numbers of stars in the FOV. The average numbers of stars in the groups were 31.4, 19.1, and 7.7 with the limiting visual magnitudes of 6, 5.5, and 4.6, respectively. The results are listed in Table 4.

It is interesting that, under Criterion A, the performance of the SVD method was close to, or even better than, the AD method, and the SVD method was more stable. However, under Criterion B, the performance of the SVD method was always worse than the AD method. This phenomenon occurred because the AD method uses the angular distance as the object of optimization estimation, so the comprehensive calibration results, considering both the camera parameters and the centroid noise, were better. It must also be mentioned that the estimation of the attitude of the star sensor was based on the star vector itself, rather than the angular distance, such that the accuracy of the camera parameters was more important; namely, Criterion A could better reflect the calibration effect than Criterion B. Therefore, the accuracy of the SVD method was as good as the AD method. The convergence rates of the two methods were also similar; as shown in Figure 6, the RMSE was calculated between the nominal angular distances and the angular distances that were obtained by the back-projection with calibrated parameters, which was the result of Group 2. Nevertheless, the elapsed time of the SVD method was significantly shorter than the AD method, especially for the case with a large number of stars in the FOV. Concerning Group 1, the calculation speed of the SVD method was improved by 94.48%, compared to the AD method.

### 4.2. Field Experiment

The SVD method is a novel on-orbit calibration method for star sensors, and therefore, it is necessary to verify the feasibility of this method by a field experiment. We tested the proposed on-orbit calibration method on the ground; the experiment platform of the field experiment is shown in Figure 7. A motorized zoom lens was adopted; the focal length was adjusted to about 16 mm; the FOV was about $19.14^{\circ }\times 11.18^{\circ }$. The resolution of the CCD is 1920×1080 and the pixel size was 2.9 μm × 2.9 μm. The experiment was carried out at the update rate of 2 Hz. To avoid the influence of the atmospheric refraction, the star sensor was placed perpendicular to the ground. The average number of observed stars in the FOV was 14.7. As the nominal centroid coordinates of observed stars were unknown, we could only use Criterion B to evaluate the performance of the field calibration experiment.

The initial values of the parameters were the same as in the simulation. The estimation results of the parameters are shown in Figure 8, where the arrows in the images represent the points of convergence, and the order of the convergence of the parameters was *f*, $k_{2}$, $k_{1}$, $u_{0}$, $u_{0}$, and then *s*. With about 2200 images, the whole estimation was convergent; $B|\mu_{\delta_{RMSE}}$ was 8.33″, and $B|\sigma_{\delta_{RMSE}}$ was 0.803″. We found that the performance in the field experiment was better than the simulation. This was because Criterion B employed the noisy simulation data, and this noise might affect the results; namely, it means the centroid errors in the field experiment were smaller than the assumption in the simulation. Therefore, the SVD method can perform well in the on-orbit calibration of a star sensor.

## 5. Conclusions

In this study, we proposed an on-orbit calibration method for star sensors by considering the fact that the singular values of the star vectors remain constant under coordinate transformation. The camera model and the principle of the SVD calibration method were presented. Using an observability analysis, the optimal calibration models for the star combinations and the singular value selection were discussed. The results of simulation with different calibration models were in accordance with our analysis, showing that the calibration model we recommended is an optimal model. Compared with the conventional AD method, the SVD method had similar accuracy; the convergence speed (the average calibration time per frame) of the SVD method, however, was significantly shorter than that of the AD method. This means the computational cost of the SVD method was smaller than the conventional methods, and so, the proposed method was more appropriate for star sensors with limited resources. Additionally, a field experiment showed that the SVD method can meet the requirements of on-orbit calibration in star sensors. Although the SVD method performed well, more tests in different situations should be conducted and further studies should be carried out, in order to learn more about this calibration method, as this is the first time it has been used for star sensor calibration. 

## Figures and Tables

**Figure 1 sensors-19-03301-f001:**
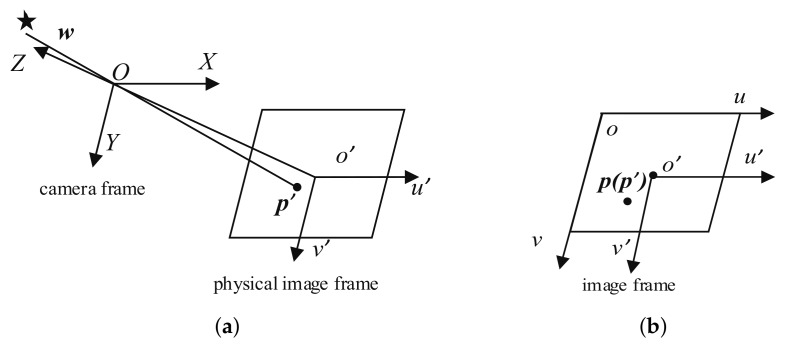
Reference frames. (**a**) Camera frame and physical image frame; and (**b**) physical image frame and image frame.

**Figure 2 sensors-19-03301-f002:**
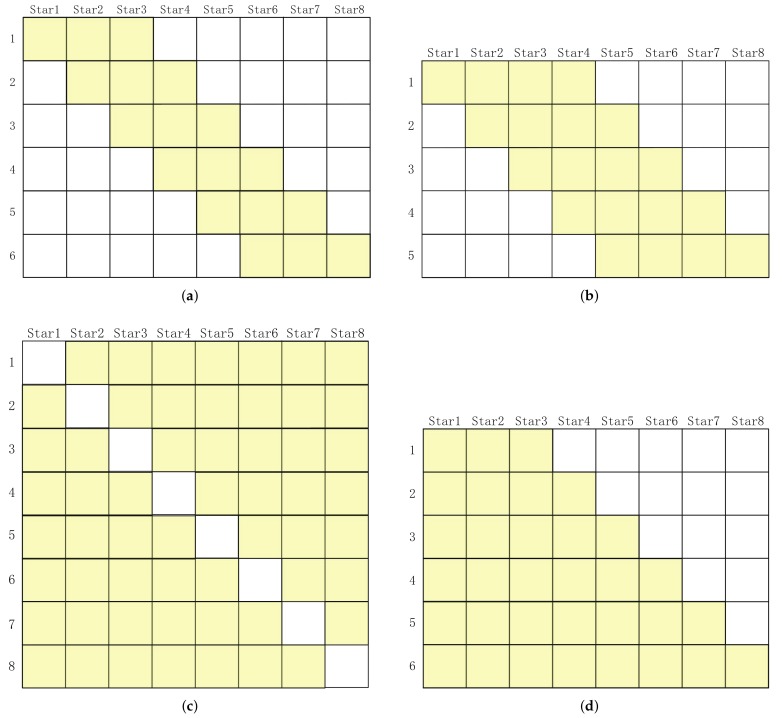
The schematic diagram of the combination models. (**a**) Model-1, (**b**) Model-2, (**c**) Model-3, and (**d**) Model-4.

**Figure 3 sensors-19-03301-f003:**
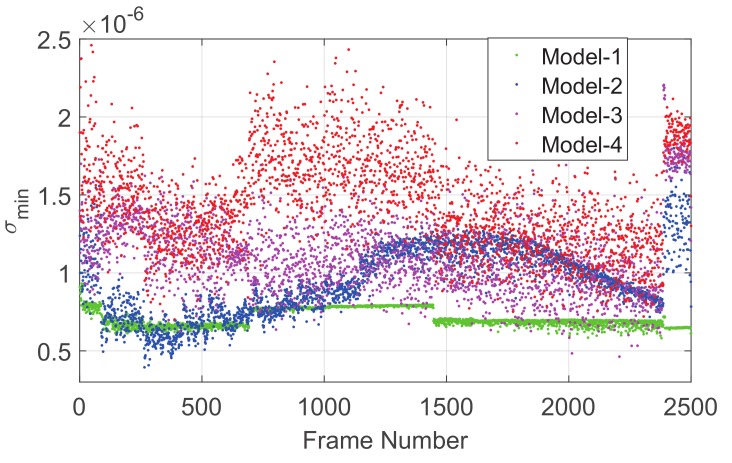
Observability analysis of the models.

**Figure 4 sensors-19-03301-f004:**
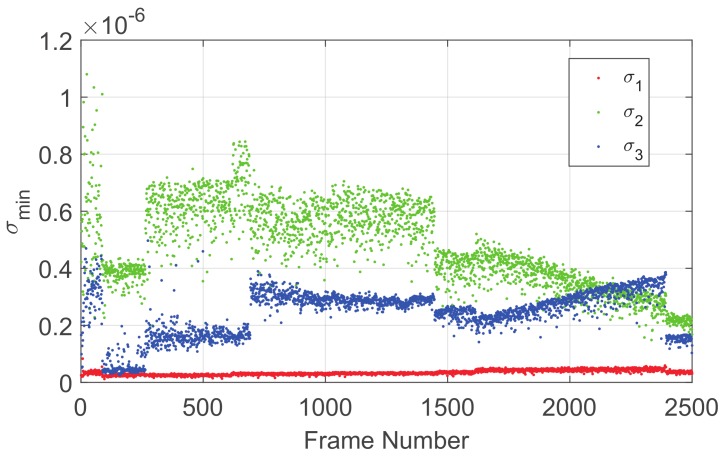
Observability analysis of the three singular values.

**Figure 5 sensors-19-03301-f005:**
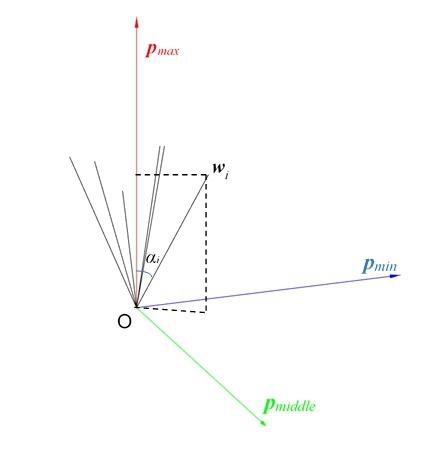
Physical interpretation of the SVD of the matrix **W**.

**Figure 6 sensors-19-03301-f006:**
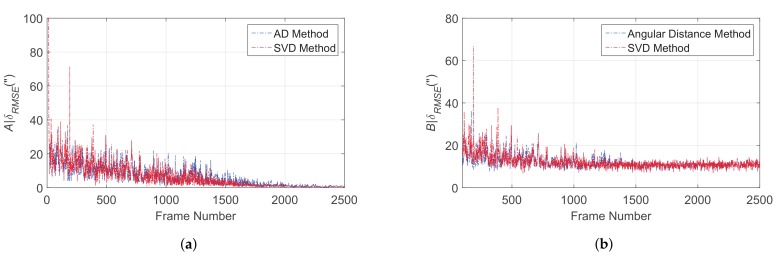
Residual errors of two calibration methods for Group 2. (**a**) Residual errors based on Criterion A and (**b**) residual errors based on Criterion B.

**Figure 7 sensors-19-03301-f007:**
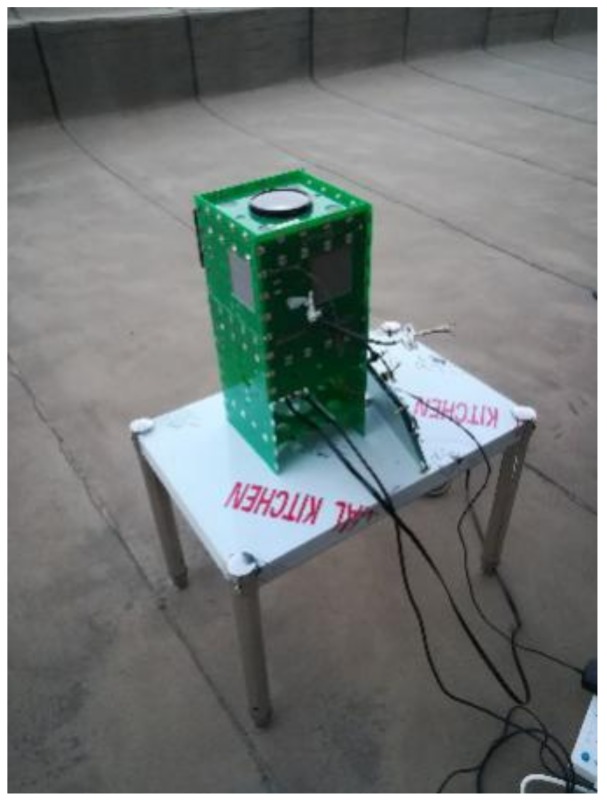
Experiment platform.

**Figure 8 sensors-19-03301-f008:**
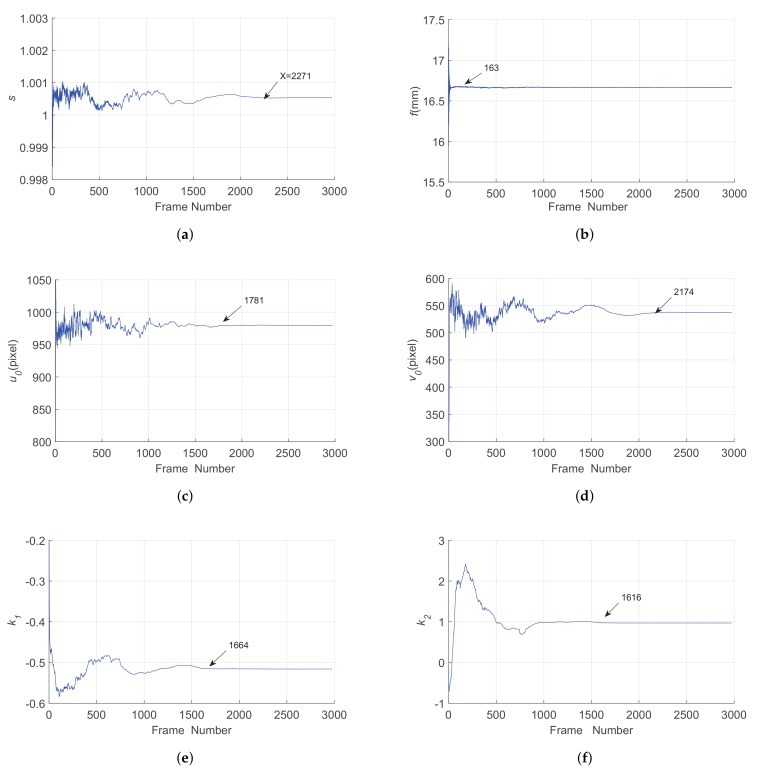
Estimation results of the parameters.(a)*s* (b)*f* (c)$u_{0}$ (d)$v_{0}$ (e)$k_{1}$ (f)$k_{2}$.

**Table 1 sensors-19-03301-t001:** The nominal values and precision of the parameters.

Parameters	*s*	*f* (mm)	$u_{0}$ (pixel)	$v_{0}$ (pixel)	$k_{1}$	$k_{2}$
nominal values	1	16	970	550	−0.5	0.5
precision	$10^{-5}$	$10^{-4}$	1	1	$10^{-2}$	$10^{-1}$

**Table 2 sensors-19-03301-t002:** Calibration results by different models.

	Model-1	Model-2	Model-3	Model-4
Δs (%)	−0.0032	0.0005	−0.0012	0.0024
Δf (%)	0.0044	0.0044	0.0050	0.0044
$\Delta u_{0}$ (%)	−2.6354	0.1126	−0.0599	−0.2159
$\Delta v_{0}$ (%)	0.9508	−0.1315	0.0940	−0.6011
$\Delta k_{1}$ (%)	−0.4379	1.1205	0.8043	1.0729
$\Delta k_{2}$ (%)	−47.4027	32.7026	15.7377	25.0508
$A|\mu_{\delta_{RMSE}}$ (")	0.852	1.106	4.392	0.465
$A|\sigma_{\delta_{RMSE}}$ (")	0.029	0.060	0.022	0.038

**Table 3 sensors-19-03301-t003:** Calibration results by different singular value combinations.

	$\left\{\sigma_{1},\sigma_{2},\sigma_{3}\right\}$	$\left\{\sigma_{2},\sigma_{3}\right\}$	$\left\{\sigma_{1},\sigma_{2}\right\}$	$\left\{\sigma_{1},\sigma_{3}\right\}$	$\left\{\sigma_{1}\right\}$	$\left\{\sigma_{2}\right\}$	$\left\{\sigma_{3}\right\}$
$A|\mu_{\delta_{RMSE}}$ (")	0.472	0.465	0.701	2.127	3.383	0.384	2.246
$A|\sigma_{\delta_{RMSE}}$ (")	0.029	0.038	0.028	0.066	0.096	0.058	0.065
time (ms/frame)	8.28	6.88	6.92	6.88	5.8	5.88	5.76

**Table 4 sensors-19-03301-t004:** Calibration results of the comparison experiment. AD, angular distance.

	Group 1		Group 2		Group 3	
Number of Stars	31.4		19.1		7.7	
**Method**	**AD**	**SVD**	**AD**	**SVD**	**AD**	**SVD**
$A|\mu_{\delta_{RMSE}}$ (")	0.535	0.436	0.419	0.465	0.344	0.244
$A|\sigma_{\delta_{RMSE}}$ (")	0.293	0.039	0.073	0.038	0.004	0.003
$B|\mu_{\delta_{RMSE}}$ (")	10.824	10.854	10.608	10.630	10.611	10.612
$B|\sigma_{\delta_{RMSE}}$ (")	0.879	0.888	1.295	1.304	2.569	2.565
time (ms/frame)	260.12	14.36	31.12	6.88	4.68	2.76

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
