# Peer review of "A Star Sensor On-Orbit Calibration Method Based on Singular Value Decomposition"

_sensors, 2019, doi:10.3390/s19153301_

Reviewer 1 Report

This paper presented a on-orbit calibration method for star sensor using SVD. I have a few questions for the authors after reading the manuscript:

For calibration, it's not clear what is the step-by-step procedure other than the description about the mathematical models.

It is not sure what calibration target and feature points are used. Also, I feel adding one scattered plot about the residual errors might be useful in visualizing the RMS errors intuitively.

I think the detailed model information of the hardware system components (e.g., camera, lens) should be given. Otherwise, it is hard to judge whether the pinhole model is a valid assumption for the star sensor

For the comparison of AD and SVD, there are sharp spike values in RMSE values, any insights on the actual causes?

The explanation of Figure 8 is not very clear for me to interpret why this result is better than simulation.

Also, I am a little confused about the term "simulation experiments" since I do not know whether this is simulation or experiments or a combination of both (it sounds like pure simulation from the paper content). However, this is only a minor suggestion.

Author Response

First, we would like to extend our appreciation for the reviewer’s thoughtful comments concerning our manuscript entitled “A Star Sensor On-orbit Calibration Method Based on Singular Value Decomposition”. Those comments are all valuable and very helpful for revising and improving our paper, as well as the important guiding significance to our researches. We have studied the comments carefully and have made revisions which we hope meet with approval. Revised portion are marked in the revised manuscript. 

Reviewer 2 Report

The paper proposes a star sensor on-orbit calibration method for the estimation of the optical parameters from which the navigation accuracy strongly depends on. The proposed method is different from the conventional on-orbit calibration methods which basically rely on the angular distance between stars. The method is based on the invariance of the singular values in coordinate transition studied by the singular value decomposition. An efficient model of the star combinations for calibration was explored by means of observability analysis. The singular value choice for calibration was carried out according to the physical interpretation of singular value decomposition of star vector matrix. The accuracy and high convergence speed were proved by the comparison between the proposed method and the conventional angular-distance-based methods. An experiment was carried out to demonstrate the applicability and feasibility of singular value decomposition even with limited computational resources.

The manuscript is well organized in several sections starting from the presentation of the camera to be calibrated, explaining the principle of the invariant singular values of star sensors and the singular value decomposition calibration method through the Extended Kalman Filter and then carrying out simulation and field experiments in order to evaluate the calibration method performances. Despite this, the reviewer has some observations and comments:

1.    the illustration of the reference frames of the camera model (line 66-75) should be improved in order to allow an easier understanding of the reader; furthermore, the reviewer suggests to insert sub-captions in Figure 1 (a-b) to facilitate the explanation of the camera model; the linewidth of the several vectors should be the same;

2.    In Eq. (4) the term s appears; it was stated the it represents the aspect ratio; could the authors give a definition and a full explanation of it?

3.    Why the actual values of the horizontal and vertical pixel sizes do not match the values given in the datasheet of the camera? (line 83);

4.    In line 112, does the acronym mean Field of View? The acronyms should be expressed in extended form at the first time used in the manuscript;

5.    why did the authors decide to exploit the Extended Kalman Filter?

6.    do other kinds of filter exist?

7.    the reviewer suggests to insert some references about Extended Kalman Filter in order to allow readers to understand how it works;

8.    from line 190 to 196, four different measurement models were proposed; do the authors think that it is possible to use other combination of stars to obtain the 3 non-zero singular values?

9.    the font size of axes in Figure 2 3 4 and 5 should be increased in order to improve the readability of the presented graphs;

10. in line 230 it was stated that a number of 2400 images can ensure the convergence of calibration; which factors influence the convergence behaviour? Is it possible to improve the convergence process?

11. why did the authors used only two kinds of criterion to test the calibration method? Do the authors think that the use of other types of criteria can lead to a better evaluation of the model performances?

12. On which base the initial values were estimated in Test1?

13. Can the authors give an explanation about the instability of σ2 obtained in performance analysis of singular values?

14. as observed previously, the label size of the axes and legend in Figure 6 and 8 is too small to be read.

Author Response

First, we would like to extend our appreciation for the reviewer’s thoughtful comments concerning our manuscript entitled “A Star Sensor On-orbit Calibration Method Based on Singular Value Decomposition”. Those comments are all valuable and very helpful for revising and improving our paper, as well as the important guiding significance to our researches. We have studied the comments carefully and have made revisions which we hope meet with approval. 

Reviewer 3 Report

This paper studies the on-orbit calibration based on singular value decomposition. In this paper, algorithms are presented, and experiments are conducted to verify the efficiency and robustness of the algorithm. However, the current version is not suitable to publish in the journal. An extensive edition is suggested. Here detail my comments:

The English should refined. There are a lot of grammatical error throughout the paper. Authors are suggest to consult a native speaker and refine the paper. For example: 

(1a). In Line 3 and 4 in page 1, These two sentences are independent, and a period instead of a comma should be used. Similar problems exist throughout the whole paper.

(1b). In Line 6, should "coordinate transition" be "coordinate transformation"?

(1c). In Line 105, should "attitude matrix" be "direct cosine matrix (DCM)"?

In Line 76, w=[x,y,z] are all small characters. In Fig. 1, however, (X,Y,Z) are in capital size. I suppose they are the same. Please unify the notation.

In Line 76, is a star vector a unit vector? If yes, please add a description.

In Eq. (1), I don't understand why authors add the 3rd row? The equation still works with the first two rows.

Do authors come up with Eqs.(1) -- (4)? If not, please add a citation.

In Line 119, Eq. (14) is a similarity transformation as long as C is orthogonal in this paper.  There has nothing to do with the definite of the matrices W and V.

In Line 118, how do authors conclude WW^T and VV^T be positive definite? Can they be positive semi-definite?

Author Response

First, we would like to extend our appreciation for the reviewer’s thoughtful comments concerning our manuscript entitled “A Star Sensor On-orbit Calibration Method Based on Singular Value Decomposition”. Those comments are all valuable and very helpful for revising and improving our paper, as well as the important guiding significance to our researches. We have studied the comments carefully and have made revisions which we hope meet with approval. Revised portion are marked in the revised manuscript. In addition, the paper has been checked for grammar, spelling, punctuation and some improvement of style by MDPI English editing. We marked in highlight in the paper. 

Round  2

Reviewer 3 Report

This paper discusses on-orbit calibration of a star sensor using SVD method. In this version, the English is refined, and problems in the previous version are corrected or clarified. I don't have further comments, and recommend this paper to publish in the journal.